# Serum SARS-CoV-2 Antigens for the Determination of COVID-19 Severity

**DOI:** 10.3390/v14081653

**Published:** 2022-07-28

**Authors:** Julien Favresse, Jean-Louis Bayart, Clara David, Constant Gillot, Grégoire Wieërs, Gatien Roussel, Guillaume Sondag, Marc Elsen, Christine Eucher, Jean-Michel Dogné, Jonathan Douxfils

**Affiliations:** 1Department of Laboratory Medicine, Clinique St-Luc Bouge, 5004 Namur, Belgium; guillaume.sondag@student.uclouvain.be (G.S.); marc.elsen@slbo.be (M.E.); christine.eucher@slbo.be (C.E.); 2Department of Pharmacy, Namur Research Institute for LIfe Sciences, University of Namur, 5000 Namur, Belgium; constant.gillot@unamur.be (C.G.); jean-michel.dogne@unamur.be (J.-M.D.); jonathan.douxfils@unamur.be (J.D.); 3Department of Laboratory Medicine, Clinique St-Pierre Ottignies, 1340 Ottignies-Louvain-la-Neuve, Belgium; jean-louis.bayart@cspo.be (J.-L.B.); gatien.roussel@cspo.be (G.R.); 4Qualiblood s.a., 5000 Namur, Belgium; clara.david@qualiblood.eu; 5Department of Internal Medicine, Clinique Saint-Pierre Ottignies, 1340 Ottignies-Louvain-la-Neuve, Belgium; gregoire.wieers@cspo.be

**Keywords:** COVID-19, SARS-CoV-2, RT-PCR, antigenic assay, prognosis test

## Abstract

The diagnostic of SARS-CoV-2 infection relies on reverse transcriptase polymerase chain reactions (RT-PCRs) performed on nasopharyngeal (NP) swabs. Nevertheless, false-negative results can be obtained with inadequate sampling procedures, making the use of other biological matrices worthy of investigation. This study aims to evaluate the kinetics of serum N antigens in severe and non-severe patients and compare the clinical performance of serum antigenic assays with NP RT-PCR. Ninety patients were included in the study and monitored for several days. Disease severity was determined according to the WHO clinical progression scale. Serum N antigen levels were measured with a chemiluminescent assay (CLIA) and the Single Molecular Array (Simoa) assay. Viremia thresholds for severity were determined and proposed. In severe patients, the peak antigen response was observed 7 days after the onset of symptoms, followed by a decline. No real peak response was observed in non-severe patients. Severity thresholds for the Simoa and the CLIA provided positive likelihood ratios of 30.0 and 10.9 for the timeframe between day 2 and day 14, respectively. Sensitive detection of N antigens in serum may thus provide a valuable new marker for COVID-19 diagnosis and evaluation of disease severity. When assessed during the first 2 weeks since the onset of symptoms, it may help in identifying patients at risk of developing severe COVID-19 to optimize better intensive care utilization.

## 1. Introduction

The diagnostic of SARS-CoV-2 infection still relies on molecular assays, with reverse transcriptase polymerase chain reactions (RT-PCRs) performed on nasopharyngeal (NP) swabs being considered as the gold standard detection method [1]. Antibody-based techniques, such as chemiluminescent immunoassays (CLIAs) or Single Molecular Array (Simoa) immunoassay have also demonstrated good correlation with RT-PCR on NP swabs, at least for cycle thresholds (Ct) below 33 [2,3]. Other matrices, such as saliva, plasma or dried blood spots, have been used in order to detect SARS-CoV-2 antigenemia [4,5,6]. Nevertheless, clinical performance depends on the detection methods. For instance, lateral flow assay (LFA) permits the provision of results within a couple of minutes, but the performance of these assays is relatively weak, with some rapid antigen detection (RAD) assays showing clinical sensitivity below 30% [7]. While almost all RT-PCR techniques show excellent sensitivity, with various methods demonstrating limits of detection (LODs) from 102 to 105 copies/mL, according to manufacturers’ package inserts and reference panels [8], the correlation between RT-PCR results from NP swabs and disease severity has been questioned [9,10,11,12]. The Ct values for a specimen vary between different kits and techniques (including target genes, primers and threshold fluorescence values), and Ct values may vary between different runs of the same kit [10,13]. Ct value also depends on the collection method of the sample and hence there may be variation in Ct values between two different samples obtained from the same person on the same day and run on the same assay [14]. In addition, concerns have been raised about the risk of false-negative results associated with the use of nasal and NP swabs, especially before symptom onset [15]. The B1.1.529 variant has also been reported to be more rapidly detected in the throat than in the nasopharynx, which raises questions about the reliability of RT-PCR testing NP swabs at the very beginning of infection with this variant of concern [16]. Thus, there is room for improvement in the diagnostics of SARS-CoV-2 infection, the “ideal” test being able to detect the presence of the virus with a similar or better sensitivity than RT-PCR performed on NP or throat swabs but with a better prognostic value regarding the severity of the disease, with more convenient access to biological specimens [17,18].

Blood samples are used to determine antibody production and adaptative immunity but are less frequently used in the diagnosis of acute infections, especially respiratory diseases, because of the focal nature of the infection or possible pre-existence of antibodies [19]. Nevertheless, antigen detection in non-respiratory fluids is still used for two respiratory bacteria: Streptococcus pneumonia and Legionella pneumophila. Interestingly, it has also been reported in SARS-CoV-1 infection [20]. Furthermore, multiple clinical manifestations suggest that SARS-CoV-2 can migrate from the lungs into the bloodstream [6,21]. This study aims at evaluating the clinical performance of two serum antigen assays for the diagnostic of SARS-CoV-2 infection and prognostic determination of disease severity.

## 2. Materials and Methods

### 2.1. Study Design and Patient Selection

All patients with documented molecular diagnosis of SARS-CoV-2 infection between April 2020 and July 2020 with at least one blood sample (i.e., venipuncture) collected within one day of an NP collection were potentially includible. One patient may have multiple blood collections after diagnosis with SARS-CoV-2. Minimal medical information required included the number of days since the onset of symptoms and sufficient information to establish the evaluation of disease severity according to the WHO clinical progression scale [22]. Reviewers of the medical records and technicians who performed the analyses were blinded and not allowed to communicate the results. Based on these criteria, a total of 92 SARS-CoV-2-infected patients were retrospectively retrieved from our laboratory biobank, representing 250 serum samples. Medical records were sufficiently documented for 90 patients. None of the patients was vaccinated against SARS-CoV-2. The two patients with insufficient clinical information (i.e., no information regarding days since symptom onset or clinical outcomes) were not included for the statistical analysis. The remaining 90 patients, for whom 243 samples were available, represented the study population. Serum samples of these patients were analyzed for evaluation of antigen kinetics (up to 30 days since symptoms). Among these 90 patients, 84 had blood samples collected between days 2 and 14 since the onset of symptoms. These 84 patients were included for the determination of a cut-off for the determination of disease severity (Figure 1). Out of the 90 patients, 11 patients were categorized as asymptomatic (WHO score of 1), 17 as having symptomatic ambulatory mild disease (WHO score of 2 and 3), 47 as having symptomatic hospitalized moderate disease (WHO scores of 4 and 5) and 15 as having symptomatic hospitalized severe disease (WHO score of 6 to 10). Twelve patients died (WHO score of 10). Obviously, the kinetics of antigens was only studied in symptomatic patients. The numbers of blood samples per patient and per disease category are presented in Appendix A. A description of the cohort and inclusion of the patients and corresponding samples in the different parts of this study is given in Figure 1.

Demographic data of the population are presented in Appendix A. Among the 90 included patients, 44 (48.9%) were women (median age = 78 years, interquartile range (IQR) = 57–89, min–max = 20–97) and 46 (51.1%) were men (median age = 77 years, IQR = 72–84, min–max = 33–94). A cohort of 71 pre-pandemic serum samples collected before February 2020 was also included to evaluate the specificity of antigen assays. The samples were retrieved from our laboratory biobank. The study protocol is available upon request and was in accordance with the Declaration of Helsinki and approved by the Medical Ethical Committee of Saint-Luc Bouge (Bouge, Belgium; approval number: B0392020000005). Written informed consent for participation was not required for this study, in accordance with the national legislation and the institutional requirements.

### 2.2. Nasopharyngeal Samples and Blood Collection

Nasopharyngeal samples were collected using eSwab liquid preservation medium tubes (Copan Italia, Brescia, Italy) and analyzed by RT-PCR without any delay. Blood samples were collected in serum gel tubes (BD SST II Advance^®^, Becton Dickinson, NJ, USA) and centrifuged for 10 min at 1740× *g* in a Sigma 3–16 KL centrifuge. Sera were stored in the laboratory serum biobank at −20 °C from the collection date. Frozen samples were thawed for 1 h at room temperature on the day of the antigen analysis. Re-thawed samples were vortexed before the analysis.

### 2.3. Antigen Assays

#### 2.3.1. Single Molecular Array

The SARS-CoV-2 nucleocapsid (N) antigen was quantified automatically in patient sera by a Single Molecular Array (Simoa) immunoassay using the Simoa HD-X analyser (Quanterix, Billerica, MA, USA). Samples were analyzed using the commercial SARS-CoV-2 N-Protein Advantage kit (item 103806), a paramagnetic microbead-based sandwich ELISA. Briefly, a diluted sample and anti-N protein antibody-coated capture beads and detector antibodies are combined for 35 min in the cuvette in the first step. The beads are then washed and a conjugate of streptavidin–ß-galactosidase is added to label the captured N protein. After washing, beads are resuspended with resorufin–ß-galactopyranozide for signal generation. Finally, beads are loaded in microarrays of femtoliter reaction wells. The fraction of bead-containing microwells counted with an enzyme is converted into “average enzymes per bead” (AEB). AEB values are converted into N protein concentration by interpolation from the calibration curve obtained by 4-parameter logistical regression fitting. The results are quantitative and expressed as pg/mL. This assay uses 8 calibrators ranging from 0 pg/mL to 200 pg/mL. Serum samples that provided results in the upper region of the calibration range were retested with a 1000-fold dilution. The LOD of the assay is 0.099 pg/mL. The within-run coefficient of variation (CV) is less than 10%. Positivity cut-offs in serum samples were not disclosed by the manufacturer.

#### 2.3.2. iFlash-2019-nCoV Assay

The SARS-CoV-2 N antigen was detected automatically in patient sera by CLIA using the iFlash 1800 automated magnetic CLIA (MCLIA) analyser (Shenzhen YHLO Biotech Co. Ltd., Shenzhen, China). Samples were analyzed using the commercial iFlash-2019-nCOV Antigen assay kit. Antigens in the sample will react with anti-2019-nCoV antibodies coated on paramagnetic particles and with anti-2019-nCoV acridinium ester-labelled conjugate to form a sandwich complex. Under a magnetic field, particles are absorbed to the wall of the reaction chamber, and unbound materials are washed away. Afterwards, the pre-trigger and trigger are added to the reaction mixture. The chemiluminescent signal is then measured in relative light units (RLUs). Results are determined via a 2-point calibration curve. The results are only qualitative and are expressed as a cut-off index (COI). Nevertheless, for the purpose of this study, we used the quantitative results provided by the analyser for calculation. The within-run CV ranges from 2.7 to 3.6% [23]. The manufacturer’s positivity cut-off is >1.0 COI. The indented use only includes the detection of antigens in NP swabs. In our study, we explored the possibility of detecting N antigens in serum (RUO setting).

### 2.4. SARS-CoV-2 Spike IgG

SARS-CoV-2 Spike IgG antibodies were quantified automatically in patient sera by a Simoa immunoassay using the Simoa HD-X analyser (Quanterix, Billerica, MA, USA). Samples were analyzed using the commercial SARS-CoV-2 Spike IgG Advantage kit (item 103769), using a similar procedure as for the SARS-CoV-2 N antigen. The positivity cut-off of 924 ng/mL corresponding to the maximal value obtained in pre-pandemic serum samples was used [24]. SARS-CoV-2 antibodies were available for 233 samples from 84 patients out of the 90 (93.3%) due to insufficient residual serum samples.

### 2.5. Reverse Transcriptase Polymerase Chain Reaction

Reverse transcriptase polymerase chain reaction for SARS-CoV-2 determination in NP swab samples was performed on a LightCycler 480 Instrument II (Roche Diagnostics, Rotkreuz, Switzerland), using the LightMix Modular SARS-CoV E-gene set (for a few samples originating from Clinique Saint-Luc Bouge), and on the GeneXpert instrument (Cepheid, Sunnyvale, CA, USA), using the Xpress SARS-CoV-2 assay targeting N2 and E genes (for the majority of samples originating from Clinique Saint-Luc Bouge and Clinique Saint-Pierre Ottignies).

### 2.6. Statistical Analyses

Descriptive statistics were used to analyze the data. Smoothing splines with four knots were used to estimate the time kinetics curves, using all longitudinal samples from the study population. Difference between disease severity per time intervals (i.e., <3 days, 4 to 10 days, 11 to 20 days and >20 days) and difference between time intervals per severity were assessed using an ordinary two-way ANOVA with Šidák’s multiple comparison tests, with individual variances computed for each comparison. Positivity cut-offs for the Simoa and the iFlash assays were determined on a previous cohort already described elsewhere [11]. Sensitivity was defined as the proportion of patients with a positive antigen test who were initially positive according to an RT-PCR SARS-CoV-2 determination in NP swab samples. Specificity was defined as the proportion of pre-pandemic samples classified as negative.

Antigen results have also been used to define cut-offs for severity. Receiver operating characteristic (ROC) curves for antigen assays were performed and the corresponding areas under the curves (AUCs) were calculated. The Youden index was used to determine the optimal severity cut-off. Additionally, a positive likelihood ratio (sensitivity/1-specificity) was calculated based on the severity cut-off for prediction of severe forms of COVID-19.

The between-patient comparison of antigen levels in the presence or in the absence of SARS-CoV-2 Spike IgG response was performed using Welch’s *t*-test. Pearson’s correlation coefficients were used to investigate the correlation between SARS-CoV-2 Spike IgG and N antigen results and between the two antigen assays.

Data analysis was performed using GraphPad Prism software (version 9.3.0, San Diego, CA, USA) and MedCalc software (version 14.8.1, Ostend, Belgium). *p* < 0.05 was used as a significance level. Our study fulfilled the ethical principles of the Declaration of Helsinki.

## 3. Results

Based on the Youden index recommended values, the following positivity cut-offs for N antigens were found for the iFlash and the Simoa assays: >0.310 COI and >0.099 pg/mL. The application of these cut-offs on the pre-pandemic cohort resulted in specificities of 95.8% (95% CI: 88.1–99.1; three false-positive results) and 98.6% (95% CI: 92.4–99.9; one false-positive result), respectively. False-positive pre-pandemic samples were different between the iFlash and the Simoa assays.

The kinetics of the N antigens in the studied population are presented in Figure 2 for both the Simoa and the iFlash assay. The peak antigen response was observed at day 7 in severe patients using both assays. Afterwards, a decline was observed up to day 20. In non-severe patients, the antigen response corresponded to a plateau phase that slowly decreased over time. The difference in kinetics between severe and non-severe patients was more distinguishable using the Simoa assay (Figure 2).

Using ROC curve analyses on samples collected in patients from day 2 to day 14, the optimal cut-off to identify severe patients was found to be 5043 pg/mL for the Simoa (AUC = 0.927 (95% CI: 0.847–1.00), sensitivity = 84.6% (95% CI: 57.8–97.3), specificity = 97.2% (95% CI: 90.3–99.5, positive likelihood ratio = 30.0) and 313.8 COI for the iFlash (AUC = 0.895 (95% CI: 0.806–0.984), sensitivity = 76.9% (95% CI: 49.7–91.8), specificity = 93.0% (95% CI: 84.6–97.0), positive likelihood ratio = 10.9) (Appendix A). The application of these severity cut-offs on kinetic models permitted us to identify the best timing since symptom onset to identify severe patients (i.e., from 4 to 10 days) (Figure 2 and Appendix A, red dotted lines). Kinetic models were also realized by matching sub-cohorts by age and gender. The results were similar, although some disproportions were observed for gender (more men in the severe group than women) (data not shown).

By splitting timing since symptoms into distinct categories (i.e., <3 days, 4 to 10 days, 11 to 20 days and >20 days), the antigen levels in severe patients observed between days 4 and 10 were significantly higher compared to non-severe patients, whatever the time category. These observations were similar for both antigen assays (Figure 3). Using the Simoa assay, the sensitivity of antigens in serum up to 10 days since symptom onset was 100% in severe patients and ranged from 91.3% (<3 days) to 100% (from 4 to 10 days) in non-severe patients (global sensitivity = 96.9%). Between days 11 and 20, the sensitivities decreased to 88.5% and 86.5% and after 20 days waned to 75.0% and 35.3% in severe and non-severe patients (Figure 3). Using the iFlash assay with the optimized positivity cut-off of 0.31 COI, the sensitivity of antigen detection in serum up to 10 days since symptom onset was 100% in severe patients and ranged from 93.5% (<3 days) to 96.0% (from 4 to 10 days) in non-severe patients (global sensitivity = 96.2%). Between days 11 and 20, the sensitivities decreased to 80.8% and 86.5% and after 20 days dampened to 75.0% and 23.5% in severe and non-severe patients.

These two antigen assays showed a highly significant correlation with a Pearson’s *r* of 0.96 (*p* < 0.0001) (Appendix A). Among the 243 samples, only 11 (4.5%) were discordant. Three results were positive for the iFlash assay (considering the optimized cut-off of 0.31 COI, though negative using the manufacturer’s cut-off of 1.00 COI; i.e., range: 0.33–0.54 COI) but negative for the Simoa, and 8 were negative for the iFlash assay but positive on the Simoa (range: 0.32–30.9 pg/mL). There was always a time window where results were concordant between the two assays considering samples collected before a mean time of 10 days (Appendix A).

In general, the mean concentration of serum antigens declined when SARS-CoV-2 Spike IgG started to be generated (Appendix A). The mean concentration of serum antigens was significantly lower in SARS-CoV-2 Spike IgG positive samples (*p*-value < 0.0001) (Figure 4A), and negative correlations between SARS-CoV-2 Spike IgG and antigen results were also identified (Pearson’s *r* of −0.59 (iFlash) to −0.65 (Simoa); *p* < 0.0001) (Figure 4B). Most patients (98.5% for the Simoa and 97.8% for the iFlash) with negative SARS-CoV-2 Spike IgG results were positive for antigens in serum using both assays (Figure 4A). Furthermore, more severe patients had lower Spike IgG compared to non-severe patients (207.5 versus 45.2 ng/mL; *p* = 0.037) (Appendix A).

## 4. Discussion

To date, few studies have investigated the detection of blood antigens of patients with SARS-CoV-2 infection [4,5,6,19,25,26]. In this study, which included 90 patients for whom 243 blood samples were obtained at different times since the onset of symptoms, we evaluated the kinetics of the SARS-CoV-2 N antigen in serum from infected patients using two antigen assays, the Simoa and the iFlash assays. Disease severity was evaluated using the WHO clinical progression scale [22].

Similar to previous studies investigating the possible prognostic value of antigen testing in serum samples [4,6,27,28,29], we evaluated whether these antigen assays could help in identifying patients more at risk of developing severe forms of COVID-19. These studies revealed that N antigen determination at the time of diagnosis, especially within the first week of symptom onset, may have potential value in triaging patients for higher-level care [26,29]. Higher concentrations of N antigens were observed in more severe patients [4,6,26,27,28,29], as well as positive correlations with inflammatory biomarker levels (i.e., CRP or IL-6) [27,28,30]. Our study confirms that severe patients exhibit higher N antigen levels compared to non-severe patients and during the first days since the onset of symptoms (Figure 2 and Figure 3). The difference between severe and non-severe patients was especially noticeable between days 4 and 10. We also estimated that cut-offs for identifying patients more at risk of severe disease were 5043 pg/mL and 313.8 COI on the Simoa and the iFlash assays. The likelihood ratios for these cut-offs were 30.0 and 10.9, which reflects their good capacity to distinguish severe from non-severe patients during the day 2 to day 14 timeframe. The likelihood ratio was lower using the iFlash assay, and the larger range of antigen concentration of the Simoa technology (from 0.099 to 100,000 pg/mL) might explain why the Simoa may be a better predictor of severe outcomes compared to the iFlash, even if a high correlation was found between the two techniques (Appendix A). Additionally, the higher sensitivity of the Simoa assay permitted us to determine earlier on which patients were more at risk of severe COVID-19, as depicted by the AUCs of the ROC analysis provided in Appendix A (Simoa AUC of 0.926 (95% CI: 0.857–0.996) compared to YHLO AUC of 0.859 (95% CI: 0.758–0.952) for the earliest samples collected in the day 2 to 14 timeframe). The measurement of S antigens in blood has received less attention, but higher concentrations in more severe patients were also observed [11,31].

Various assays (Simoa, Lumipulse G, Quantigene, MesoScale and Biohit healthcare assays) revealed a specificity ranging from 97.0% to 100% on pre-pandemic samples [6,11,19,28,29,31]. Ogata et al. were the first to report on the Simoa technology and found a specificity of 82.4% [4]. The method they used was, however, home-made and does not represent the current performance of the Simoa assay. As a matter of fact, Shan et al. found a specificity of 100% using the commercial kit from Quanterix on the Simoa [6]. In our evaluation, the specificities of the iFlash and Simoa were 95.8% and 98.6% and were therefore consistent with the data from the literature.

The clinical performance of these antigen assays was also directly compared to RT-PCRs performed on NP swabs [6,11,12,19,28,29]. The clinical performance was directly related to the design of published studies. In samples collected within 12 h of NP sampling, we previously found that the clinical sensitivity and specificity of the Simoa assay were 100% and 92.3%, respectively [12]. Using the same cohort, the clinical performance of the iFlash assay was the same (data not shown).

Considering studies that included patients who developed symptoms up to a maximum of 2 weeks, the clinical sensitivity ranged from 85.2% to 93.0% [19,26,29] and increased to 94.2% to 100% with samples collected within the first days since symptom onset [11,12,19,27]. However, clinical sensitivity significantly decreased after 2 weeks (43.2% to 74.5%) [6,19,26,29] and >4 weeks since symptom onset (from 0% to 34.2%) [26,32]. Longitudinal monitoring of antigen concentrations more than 2 weeks after the onset of symptoms is therefore not likely to be helpful in predicting outcomes or responses to therapy [26]. Six studies found moderate clinical sensitivities ranging from 41.0% to 74.0% [4,25,27,28,30,33]. This lower performance is explained by the design of these studies. Indeed, the time since symptom onset was either not disclosed [25,27,28,33] or was long (i.e., up to 30 days), decreasing the probability of having positive samples given the kinetics of the SARS-CoV-2 N antigen in the blood [4,30]. Given that the peak of the N antigen is reached after 7 days, as for the viral load in NP samples [34], and that a continuous decline is observed afterwards, the timing since symptoms is of paramount information for the evaluation of antigenemia (Figure 2). In our evaluation, we confirmed that the clinical sensitivity of the N antigen test as compared to RT-PCR was excellent (96.2% for iFlash and 96.9% for Simoa) in samples collected earlier on following symptom onset (i.e., <10 days), especially in more severe patients (i.e., 100% for both assays) (Figure 3), as previously reported [26,29]. Of note, the performance of the S antigen assay (sensitivities of 64% to 85.3%) in blood was lower compared that of the N antigen assay (89.0% to 100%) using the same cohorts of patients [11,31]. This is probably due to the larger copy number of N proteins per viral particle (i.e., around 1000) [6].

Interestingly, patients who exhibited a level of SARS-CoV-2 Spike IgG above the positivity cut-off showed statistically significantly lower serum antigen levels (Figure 4). As expected from previous studies in COVID-19 subjects, the level of antibodies started to increase approximately 10 days after the onset of symptoms [11,35]. It also correlates with the waning of N antigen levels, as observed in Figure 4, and in another study measuring anti-NCP antibodies (which are positively correlated with anti-S antibodies [26,36]. These data support the concept that antigenemia may be a prognostic marker of severity to identify patients more likely to require intensive care in the first few days after the onset of symptoms.

In the literature, the clinical specificity of N antigen assays has been reported to range from 68.0% to 99.8% (mean clinical specificity = 90.9%) [4,25,26,29,32,33]. This implies that some patients test positive for N antigens in blood but negative for NP RT-PCR. It should be noted that this could be due to false-negative NP RT-PCR results in infected patients. In the study of Le Hingrat et al., 8 out of 12 patients with a negative RT-PCR result had detectable N antigens in blood. In six of these eight patients (i.e., 75%), RT-PCR performed on lower respiratory tract samples turned out to be positive, showing that the virus may have migrated from the nasopharyngeal sphere to the lower respiratory tract [19]. Additionally, Su et al. found that 16 (32.2%) samples from 50 COVID-19-negative patients by pharyngeal RT-PCR were positive for the presence of N antigens, including two severe COVID-19 patients [25]. The detection of blood antigens can therefore be a valuable alternative to RT-PCR, especially in cases of negative results. The application of antigen testing in blood compared to RT-PCR should be validated locally and depend on laboratory settings (cost, access, etc.). Antigen testing could also be used for large-scale screening using dried blood spots, for example.

In a previous study, the application of ROC curve adapted cut-offs (1.85 to 10 pg/mL) allowed a significant increase in clinical sensitivity (from 76 to 92%), with a limited impact on specificity (from 100% to 96.84%) [33]. In our study, we increased the clinical performance of iFlash (1.0 to 0.31 COI). This approach has also been successfully used for SARS-CoV-2 antibody assays [37].

The impact of variants of concern (VOCs) on the performance of the two antigen assays used in our study was not studied. Even if the use of N antigen assays might mitigate the impact of VOCs, this should be verified in further studies.

## 5. Conclusions

Sensitive N antigen detection in serum provides a valuable new marker test for COVID-19 diagnosis, only requiring a blood draw, that is scalable in all clinical laboratories. It allows potential new developments of design rapid antigen blood tests or combined ELISA assays for the detection of both antigens and antibodies. Measuring antigens in blood presents several advantages, including a more standardized process of obtaining blood samples compared to RT-PCR, with the possible development of dry blood spot sampling strategies. Importantly, antigenemia, when assessed during the first 2 weeks since the onset of symptoms, may help in identifying patients at risk of developing severe COVID-19. The severity cut-offs proposed in our investigation need to be confirmed in subsequent studies, but our results already support the conclusion that, compared to the NP RT-PCR, the detection of N antigens in blood allows better discrimination of severe and non-severe cases of COVID-19. In addition, these techniques are at least as accurate as NP RT-PCR for diagnosing SARS-CoV-2 infection, creating the possibility of developing more convenient testing strategies for patients. They may finally facilitate patient triage to optimize better intensive care utilization in patients presenting early after symptom onset.

## Figures and Tables

**Figure 1 viruses-14-01653-f001:**
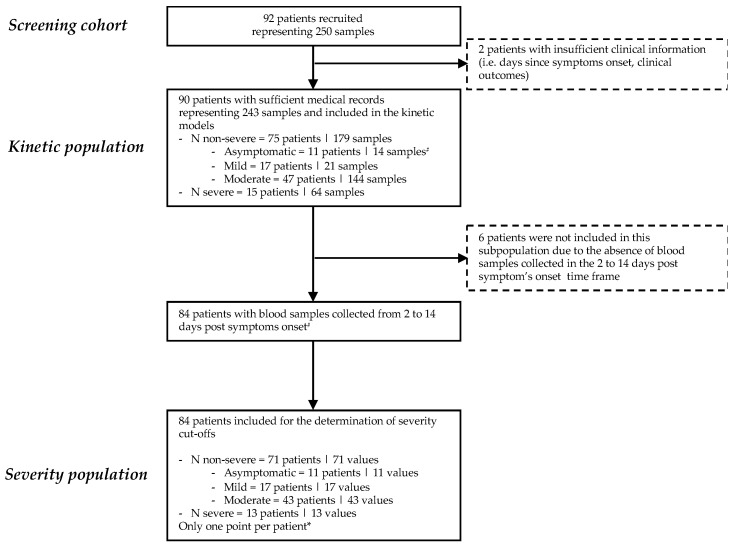
Study flow diagram. ^#^ In some kinetic models, asymptomatic patients and patients with positive SARS-CoV-2 spike immunoglobulin G were excluded. * Multiple sample inclusion criteria were applied to investigate different possible cut-offs for severity: (a) earliest antigenemia value since symptom onset within the day 2 to day 14 window for a particular patient; (b) mean of all antigenemia values of samples collected within the day 2 to day 14 window for a particular patient; and (c) the maximal antigenemia value obtained within the day 2 to day 14 window. All these inclusion strategies reported one value per patient (see Appendix A).

**Figure 2 viruses-14-01653-f002:**
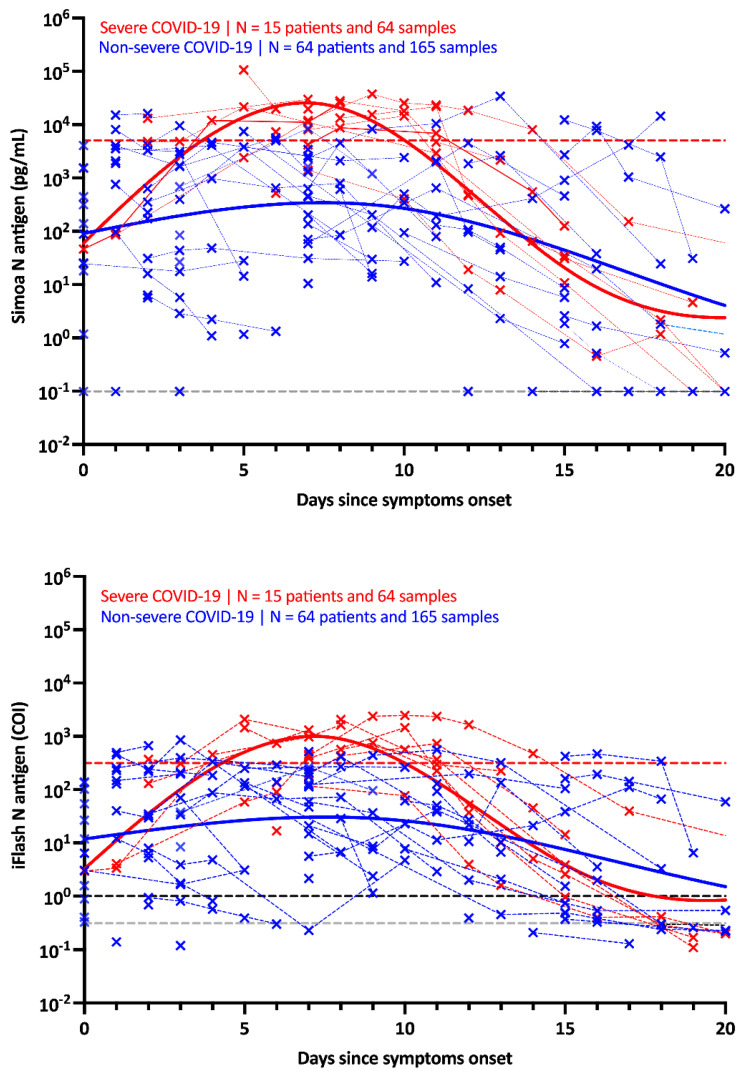
Kinetics of antigenemia since the onset of symptoms in non-severe and severe patients. The grey dotted lines correspond to the positivity cut-off of each antigen assay, as found by ROC curve analyses. The black dotted line corresponds to the positivity cut-off of the iFlash assay, as declared by the manufacturer. The red dotted lines correspond to the severity cut-off of each antigen assay, as found by the ROC curve analyses for the day 2–day 14 window. Only patients with symptoms and negative for SARS-CoV-2 Spike IgG directed against the spike protein were included in this kinetics representation.

**Figure 3 viruses-14-01653-f003:**
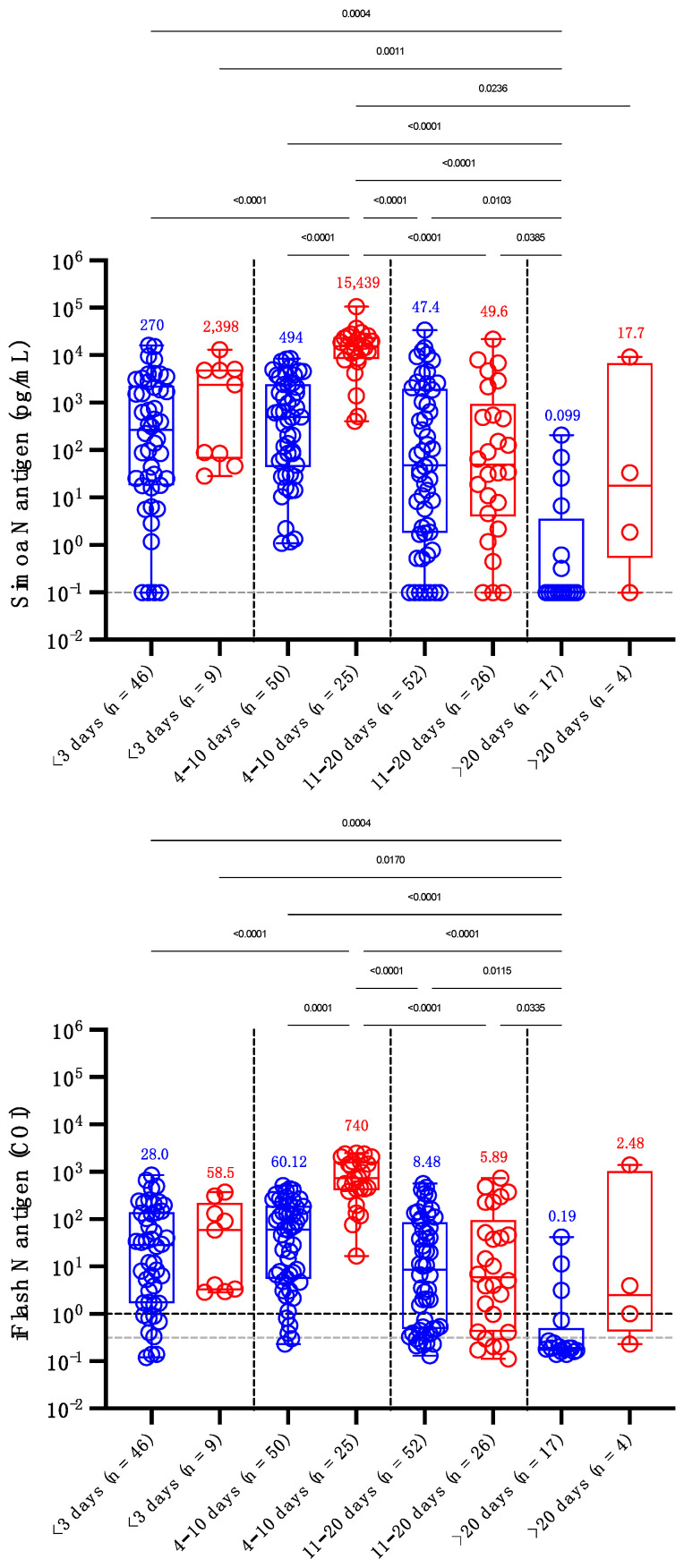
Levels of serum antigen according to the delay since the onset of symptoms (<3, 4–10, 11–20 and >20 days) and severity. Blue dots correspond to non-severe patients and red dots to severe patients. The grey dotted lines correspond to the positivity cut-off of each antigen assay, as found by ROC curve analyses. The black dotted line corresponds to the positivity cut-off of the iFlash assay, as declared by the manufacturer. Medians are represented on top of each whisker box. Only *p*-values < 0.05 are represented.

**Figure 4 viruses-14-01653-f004:**
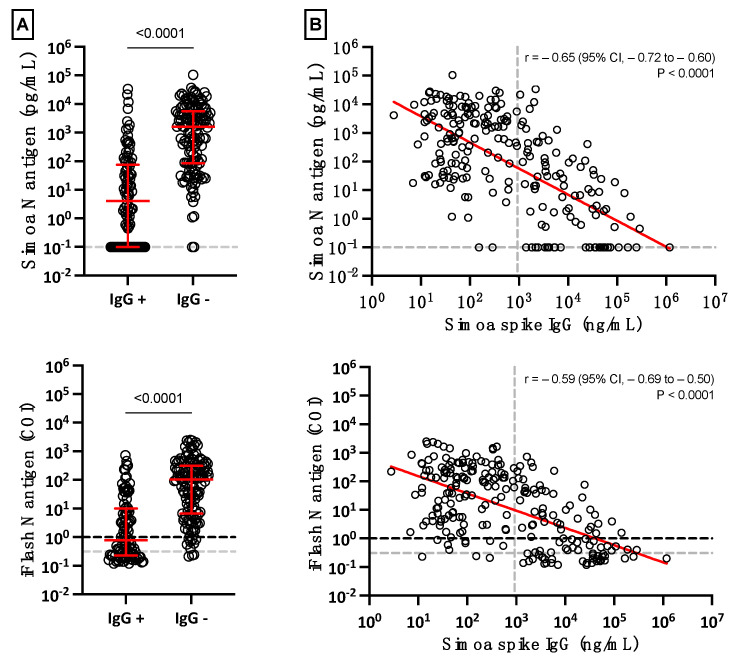
(**A**) Positive and negative SARS-CoV-2 Spike IgG results in serum according to antigenemia. The grey dotted lines on the Y-axis correspond to the positivity cut-off of each antigen assay, as found by ROC curve analyses. The black dotted line on the Y-axis of the iFlash panel represents the cut-off specified by the manufacturer. (**B**) Linear regression of antigenemia obtained with the two antigen assays versus the amount of SARS-CoV-2 Spike IgG. The grey dotted line on the X-axis represents the positivity cut-off for IgG directed against the spike protein.

## Data Availability

The data presented in this study are available on request from the corresponding author. The data are not publicly available according to the ethical committee decision on the conduct of this study.

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
