# Peer review of "Serum SARS-CoV-2 Antigens for the Determination of COVID-19 Severity"

_viruses, 2022, doi:10.3390/v14081653_

Round 1

Reviewer 1 Report

The authors evaluated the kinetics of SARS-CoV-2 N antigen level in sera of infected individuals (including both severe and non-severe infections) using two distinct assays, and observed a significantly higher level of N antigen in severe patients at day 4-10 since the onset of the symptoms. This finding is interesting and be of interest for the scientific community. However, the current manuscript suffers from some flaws.

1. The authors proposed that assessment of the antigenemia might facilitate identifying the patients with severe infections. However, when is the severe infection? Maybe at 4-10 days after the onset of symptoms when the N antigen reaches the peak, the patients have already developed the severe infection clinically. Therefore, authors should rephrase this kind of statement and avoid over-statement.

2. It is not surprised to see the correlation between N antigen level and the severity of infection. More severity form of disease is reasonably leading to a higher level of antigen protein in the serum. However, is this correlation N antigen-specific or ubiquitously existed for other viral antigens? Thus, the SARS-CoV-2 S-protein level should be examined and the correlation between S protein level in serum and the severity of infection could also be evaluated as what have done for the N antigen.

3. In Figure 3, the authors compared the antigen level of severe patients (red dots, 4-10 days) with all other groups of samples. However, what are those p values between severe patients and non-severe patients for <3 days, 11-20 days, and >20 days? Please provide these p values.

4. In Figure 4, negative correlation between anti-S IgG and N antigen level in serum samples. Are those IgG- samples more likely to be the severe patients? To echo the findings in Figure 2 and 3, the authors should also label the severe and non-severe patients in Figure 4 or performed a more systematic analysis involving all three factors, including antigen level, severity of the patients, and the anti-S level.

5. I suggest the authors to show the exact data as Figure 2. Kinetic models were also realized by matching sub-cohorts by age and gender. Results were similar although some disproportions were observed for gender (more men in the severe group than women) (data not shown).

Author Response

Reviewer 1:

The authors evaluated the kinetics of SARS-CoV-2 N antigen level in sera of infected individuals (including both severe and non-severe infections) using two distinct assays, and observed a significantly higher level of N antigen in severe patients at day 4-10 since the onset of the symptoms. This finding is interesting and be of interest for the scientific community. However, the current manuscript suffers from some flaws.

  1. The authors proposed that assessment of the antigenemia might facilitate identifying the patients with severe infections. However, when is the severe infection? Maybe at 4-10 days after the onset of symptoms when the N antigen reaches the peak, the patients have already developed the severe infection clinically. Therefore, authors should rephrase this kind of statement and avoid over-statement.

 We agree with the reviewer that the timing since symptoms is primordial to possibly identify more severe patients. The conclusion of the paper has been adapted.

“In addition, these techniques are at least as accurate than NP RT-PCR for diagnosing SARS-CoV-2 infection, giving us the possibility to develop more convenient testing strategies for patients. It may finally facilitate patient triage to optimize better intensive care utilization in patients presenting early since symptoms.”

  1. It is not surprised to see the correlation between N antigen level and the severity of infection. More severity form of disease is reasonably leading to a higher level of antigen protein in the serum. However, is this correlation N antigen-specific or ubiquitously existed for other viral antigens? Thus, the SARS-CoV-2 S-protein level should be examined and the correlation between S protein level in serum and the severity of infection could also be evaluated as what have done for the N antigen.

 The correlation between the S-protein and the severity of the disease has also been identified by our study group in another publication (https://www.degruyter.com/document/doi/10.1515/cclm-2021-1244/html). This reference is mentioned page 10, line 307-308. The performance of the S antigen assay was lower compared to N antigen assay as compared to RT-PCR (85.3% sensitivity for S compared to 100% sensitivity for N; https://www.degruyter.com/document/doi/10.1515/cclm-2021-1244/html). This probably due to the larger copy number of N proteins per viral particle (i.e. around 1,000).

  1. In Figure 3, the authors compared the antigen level of severe patients (red dots, 4-10 days) with all other groups of samples. However, what are those p values between severe patients and non-severe patients for <3 days, 11-20 days, and >20 days? Please provide these p values.

 As requested by the reviewer, all significant p values (i.e., <0.05) are now presented in the Figure 3, for both the Simoa and the iFlash assays.

  1. In Figure 4, negative correlation between anti-S IgG and N antigen level in serum samples. Are those IgG- samples more likely to be the severe patients? To echo the findings in Figure 2 and 3, the authors should also label the severe and non-severe patients in Figure 4 or performed a more systematic analysis involving all three factors, including antigen level, severity of the patients, and the anti-S level.

 As asked by the reviewer, we also studied the levels of IgG according to severity. It appeared that severe patients had significantly less IgG compared to non-severe patients (207.5 versus 45.2 ng/mL, respectively with a p value of 0.037). These results were expected since we know that severe patients had higher antigen levels and that there is an inverse relationship between IgG levels and antigenemia. The manuscript has been adapted in this way and a Supplemental Figure (number 5) has been created. However, as observed on this figure, we can see that the IgG level is less discriminating than N antigenemia for determination of severity in infected patients.

  1. I suggest the authors to show the exact data as Figure 2. Kinetic models were also realized by matching sub-cohorts by age and gender. Results were similar although some disproportions were observed for gender (more men in the severe group than women) (data not shown).

 We rather prefer not to add additional Figures since this does not add any relevant information’s for the message convey by this article. However, we provide for this reviewer the computation we have done for this analysis (please find the kinetic models below). In addition, determinants of high viral load are currently under investigation and are part of one of our other project thus, we prefer not to add too much information on this point at this stage which will necessitate integration of multivariable analyses (see document attached).

Reviewer 2 Report

General comment

The manuscript “Serum SARS-CoV-2 antigens for the determination of COVID-19 severity” aims to correlate the antigenemia and disease severity, which is interesting and inspiring. The authors presented two distinct methods and drew the same conclusion that the N antigen of SARS-CoV-2 in blood sample can be predictive for COVID-19 severity. This may be valuable for clinical diagnostic and suggestive for health care. However, I have some concerns about it.

1.      Regarding the collection of the samples how do the authors define the onset of the symptoms for the asymptomatic donors? From the study design the serum samples were collected from day 2 after the onset of the disease but this is not possible for the asymptomatic donors. The authors should explain this. It is also not clear for how long they take the serum since the data in Fig. 2 and 3 are longer than 14.

2.      The authors claimed the sensitivity of antigenemia methods were better than RT-PCR, avoiding false negative. Their hypothesis is that if the virus is in the later stage it migrates to the lower respiratory tract then RT-PCR would turn out negative if NP swabs are tested. In the current study RT-PCR were performed only with the first data sample, it might have been interesting to test samples at later time points to assess their hypothesis. The RT-PCR kinetic should be correlated to the antigenemia kinetics.  

3.      In the discussion the authors propose the detection of N antigen in blood as alternative to RT-PCR. Could this method be applied for large scale screening using blood spots for example? The authors comment this point in the discussion regarding the feasibility of replacing RT-PCR e.g. cost, time, access.

4.      Do the authors have any data about the transmissibility of individuals with high concentration of N antigen in serum compared with those with low antigen in serum?

5.      The data in the paper are based on Wuhan SARS-CoV-2 virus, I wonder if the new variants were investigated regarding the kinetic of antigenemia. Are the positive and negative cut-offs different?

6.      As far as I can see, the severity of disease is obvious upon onset of symptoms. However, as the samples presented in the paper are collected after the symptom onset it is not clear whether detection of serum N antigen is able to identify or predict the risk of severe COVID19. The author should comment this.

7.      Authors should explain more about the virus dynamics in blood. When do viruses start migrating to blood after infection? Or is there a threshold of viral load in NP for blood transferring?

Specific comments

1.      How reliable of the summary red/blue lines in figure 2? How can you interpret a sample above severity cut-off when it is from a non-severe COVID-19 patient? It would help to demonstrate the figure with a r2 or p value.

2.      In Figure 3 the pairwise significant analysis is enough to make the point that the severe patients have higher antigen in blood during the disease course. It is quite confusing to list all the significant multiple comparisons.

3.      Supplemental Figure 2: Please make the legend clear. a), b) and c) are not indicated in the figure.

4.      In Material and Methods: it is not clear from where the blood samples were collected e.g vein blood? Until which days the samples were collected?

5.      From the statistical analysis it is also not clear how the authors calculate the specificity and sensitivity.

Author Response

Reviewer 2:

The manuscript “Serum SARS-CoV-2 antigens for the determination of COVID-19 severity” aims to correlate the antigenemia and disease severity, which is interesting and inspiring. The authors presented two distinct methods and drew the same conclusion that the N antigen of SARS-CoV-2 in blood sample can be predictive for COVID-19 severity. This may be valuable for clinical diagnostic and suggestive for health care. However, I have some concerns about it.

  1. Regarding the collection of the samples how do the authors define the onset of the symptoms for the asymptomatic donors? From the study design the serum samples were collected from day 2 after the onset of the disease but this is not possible for the asymptomatic donors. The authors should explain this. It is also not clear for how long they take the serum since the data in Fig. 2 and 3 are longer than 14.

 Good point. Obviously, it is not possible to describe any delay since symptoms in asymptomatic patients. As mentioned in the manuscript: “In some kinetic models, asymptomatic patients were excluded” From the total of 90 patients, only 79 (90 – 11 asymptomatic patients) were used to perform the kinetics since symptoms). This has been better described in the paper.

 The maximum delay since symptoms was 30 days (this is now stipulated in the text).

“Serum samples of these patients were analyzed for evaluation of the antigen kinetics (up to 30 days since symptoms).”

  1. The authors claimed the sensitivity of antigenemia methods were better than RT-PCR, avoiding false negative. Their hypothesis is that if the virus is in the later stage it migrates to the lower respiratory tract then RT-PCR would turn out negative if NP swabs are tested. In the current study RT-PCR were performed only with the first data sample, it might have been interesting to test samples at later time points to assess their hypothesis. The RT-PCR kinetic should be correlated to the antigenemia kinetics.

 We agree with the reviewer that this might be very interesting. The fact that RT-PCR might be negative in case of positive antigen tests was explain in the discussion part mostly based on the available literature. However, the design of our study did not allow us to study this hypothesis.

  1. In the discussion the authors propose the detection of N antigen in blood as alternative to RT-PCR. Could this method be applied for large scale screening using blood spots for example? The authors comment this point in the discussion regarding the feasibility of replacing RT-PCR e.g. cost, time, access.

 Yes, technically, this is possible. However, it depends on the laboratory settings. Cost of RT-PCR varied widely across manufacturers. The TAT for RT-PCR also vary widely. Thus, this approach should be validated locally. Dried blood spot could also be used to facilitate the sampling, both in the clinical and the ambulatory setting.

  1. Do the authors have any data about the transmissibility of individuals with high concentration of N antigen in serum compared with those with low antigen in serum?

 We agree with the reviewer that this might be very interesting. However, the design of our study did not allow us to study this hypothesis. Nevertheless, the fact that high antigen fairly correlates with low RT-PCR Ct (see: https://pubmed.ncbi.nlm.nih.gov/35074508/) suggest that the RT-PCR recommendations on transmissibility might be applicable to antigen testing. Nevertheless, we prefer to avoid being too speculative in the manuscript and this point will therefore not further discuss.

  1. The data in the paper are based on Wuhan SARS-CoV-2 virus, I wonder if the new variants were investigated regarding the kinetic of antigenemia. Are the positive and negative cut-offs different?

 Unfortunately, we were not yet able to evaluate the impact of the different SARS-CoV-2 variants. This is the scope of our next research. Nevertheless, the fact that the antigen used in assays was the nucleocapsid make us think that there is potentially not impact of the variant (since mutations mainly have an impact in the spike protein of the virus).

  1. As far as I can see, the severity of disease is obvious upon onset of symptoms. However, as the samples presented in the paper are collected after the symptom onset it is not clear whether detection of serum N antigen is able to identify or predict the risk of severe COVID19. The author should comment this.

 This is an important point raised by the reviewer. In our ROC curves analyses (see Supplemental Figure 2), we determine the possibility to detect severe patients at the time of diagnosis (green curve – Earliest D2-D14). Importantly, thanks to its higher sensitivity, the Simoa assay permits to keep an excellent AUC (0.926 [95%CI: 0.857-0.996) compared to the less sensitive YHLO assay (AUC 0.859 [95%CI: 0.758-0.952]). In that analysis, the sample is collected at the time of diagnosis (i.e. at the time of the RT-PCR). The AUC observed clearly demonstrates that this strategy may be helpful to identify patients at risk of severe COVID-19. This discussion has been adapted accordingly:

“Also, the higher sensitivity of the Simoa assay permit to determine earlier which patient is more at risk of severe COVID-19 as depicted by the AUC of the ROC analysis provided in Supplementary Figure 2 (Simoa AUC of 0.926 [95%CI: 0.857-0.996) compared to YHLO AUC of 0.859 [95%CI: 0.758-0.952] for the earliest samples collected in the day 2 to 14 timeframe).”

  1. Authors should explain more about the virus dynamics in blood. When do viruses start migrating to blood after infection? Or is there a threshold of viral load in NP for blood transferring?

 We agree with the reviewer that this might be very interesting. However, the design of our study did not allow us to study this hypothesis. This would need more experiments, probably at the individual level, necessiting the collection of several NP swabs, bronchoalveolar fluids and blood sampling since the onset of symptoms. On an ethical point of view, if this is not part of the patient’s standard requirement during its hospitalization, it is unlikely that we get the approval for bronchoalveolar fluids since the onset of symptoms, especially in non-severe patients.

Specific comments

  1. How reliable of the summary red/blue lines in figure 2? How can you interpret a sample above severity cut-off when it is from a non-severe COVID-19 patient? It would help to demonstrate the figure with a r2 or p value.

 In Figures 2 and 3, we can observe that the difference between severe and non-severe is mostly present between days 4 and 10. Here what is mentioned in the text: “Using ROC curves analyses on samples collected in patients from day 2 to day 14, the optimal cut-off to identify severe patients have been found to be 5,043 pg/mL for the Simoa (AUC = 0.927 [95%CI: 0.847-1.00], sensitivity = 84.6% [95%CI: 57.8-97.3], specificity = 97.2% [95%CI: 90.3-99.5, positive likelihood ratio = 30.0] and 313.8 COI for the iFlash (AUC = 0.895 [95%CI: 0.806-0.984], sensitivity = 76.9% [95%CI: 49.7-91.8], specificity = 93.0% [95%CI: 84.6-97.0], positive likelihood ratio = 10.9) (Supplemental Figure 2)”. Therefore, there is still a risk of having a non-severe patient above the severity cut-off but this risk was limited in our study. Residual plots also shows that the fitting is good, especially for severe patients in the day 2 to 10 timeframe.

Residual plots:

  1. In Figure 3 the pairwise significant analysis is enough to make the point that the severe patients have higher antigen in blood during the disease course. It is quite confusing to list all the significant multiple comparisons.

 We agree that this might be confusing for some readers but we also think that this is important to evaluate the difference between groups. Additionally, another reviewer asked us to show all p value in figure 3.

  1. Supplemental Figure 2: Please make the legend clear. a), b) and c) are not indicated in the figure.

 The letters correspond to the three colors used. The legend now better described this.

  1. In Material and Methods: it is not clear from where the blood samples were collected e.g vein blood? Until which days the samples were collected?

 Vein blood indeed. This has been added in the manuscript.

  1. From the statistical analysis it is also not clear how the authors calculate the specificity and sensitivity.

 More details have been provided in the manuscript (Material and methods).

Round 2

Reviewer 1 Report

The authors have responded productively to the reviewer concerns. I recommend publication.